# Fundamental investigations on the ionic transport and thermodynamic properties of non-aqueous potassium-ion electrolytes

Shobhan Dhir [1,2], Ben Jagger [1,2], Alen Maguire[1] & Mauro Pasta [1] ✉

Non-aqueous potassium-ion batteries (KIBs) represent a promising complementary technology to lithium-ion batteries due to the availability and low cost of potassium. Moreover, the lower charge density of $K^+$ compared to $Li^+$ favours the ion-transport properties in liquid electrolyte solutions, thus, making KIBs potentially capable of improved rate capability and low-temperature performance. However, a comprehensive study of the ionic transport and thermodynamic properties of non-aqueous K-ion electrolyte solutions is not available. Here we report the full characterisation of the ionic transport and thermodynamic properties of a model non-aqueous K-ion electrolyte solution system comprising potassium bis(fluorosulfonyl)imide (KFSI) salt and 1,2-dimethoxyethane (DME) solvent and compare it with its Li-ion equivalent (i.e., LiFSI:DME), over the concentration range 0.25–2 molal. Using tailored K metal electrodes, we demonstrate that KFSI:DME electrolyte solutions show higher salt diffusion coefficients and cation transference numbers than LiFSI:DME solutions. Finally, via Doyle-Fuller-Newman (DFN) simulations, we investigate the K-ion and Li-ion storage properties for K‖graphite and Li‖graphite cells.

The lithium price has increased more than sevenfold since the start of 2021 (as of May 2022), reaching unprecedented price levels and demonstrating significant challenges for the lithium-ion battery (LIB) supply chain[1,2]. With forecasts showing a potential significant lithium supply deficit by 2030[1], the case for alternative chemistries based on abundant minerals which can fulfil some LIB functions has never been stronger[3–8]. Potassium-ion batteries (KIBs) have a significant advantage over sodium-ion batteries (NIBs) as $K^+$ can reversibly intercalate into the graphite electrodes used in LIBs[9,10], thus one of the primary components of KIBs is already available at commercial scale, unlike for NIBs[3].

Fast charging rates (~ 4C[11]) are also becoming increasingly important for batteries, particularly in electric vehicles (EVs), however, conventional LIBs are inherently limited in their rate capability[12]. KIBs, however, may have an advantage over LIBs in terms of rate and power. Early data show improved rate performance of KIBs compared to LIBs[13], suggesting faster transport in non-aqueous potassium-ion (K-ion) electrolytes[14]. The larger size of $K^+$ compared to $Li^+$ results in a lower charge density, and thus weaker interactions with solvent molecules and a smaller Stokes radius[15], which is expected to facilitate faster ionic transport in the electrolyte[3]. This is supported by the results from Landesfeind et al. who found improved transport properties of a non-aqueous sodium-ion (Na-ion) electrolyte compared to the lithium-ion (Li-ion) equivalent[16]. It also appears from various data sources collated by Landesfeind et al. that aqueous K-ion electrolytes may show faster transport properties compared to Li- and Na-ion[16]. $K^+$ has also been found to have the lowest desolvation activation energy of the three cations[14,17].

Though studies suggest improved rate performance, a comprehensive understanding of mass transport in K-ion electrolytes can only be obtained through full and accurate characterisation of the

[1]Department of Materials, University of Oxford, Oxford OX1 3PH, UK. [2]These authors contributed equally: Shobhan Dhir, Ben Jagger.
✉e-mail: mauro.pasta@materials.ox.ac.uk

fundamental ionic transport and thermodynamic properties: the salt diffusion coefficient ($D$), the cation transference number ($t_+^0$), the ionic conductivity ($\kappa$), and the thermodynamic factor ($\chi_M$)[18].

There are a growing number of studies which have fully characterised these properties for Li-ion electrolytes using a variety of electrochemical techniques[19–25]. Spectroscopic techniques, capable of determining these properties for Li-ion electrolytes through direct visualisation of concentration gradients, have also recently been developed. These include X-ray spectroscopy[26], Raman spectroscopy[27], or magnetic resonance imaging (MRI)[28,29]. However, there is currently no study which has fully characterised the ionic transport and thermodynamic properties ($D$, $t_+^0$, $\kappa$ and $\chi_M$) for a K-ion electrolyte. Given these properties are challenging and time-consuming to obtain for Li-ion electrolytes, the added complication of the extreme reactivity of K metal provides significant additional challenges to K-ion electrolyte characterisation.

In this study we comprehensively characterise the critical transport and thermodynamic properties of non-aqueous K-ion electrolyte solutions (with various concentrations) comprising of potassium bis(fluorosulfonyl)imide (KFSI) salt and 1,2-dimethoxyethane (DME) solvent and comparing them with Li-ion equivalent electrolyte solutions (LiFSI in DME) over the concentration range 0.25–2 molal. We developed a K metal preparation protocol to ensure sufficient stability and data reproducibility to enable K-ion electrolyte characterisation using the most accurate electrochemical characterisation techniques. FSI⁻ in DME is used as a model electrolyte system because of the ability of the FSI⁻ anion to more effectively passivate the K and Li metal surface[30–32] and due to the stability of ethers in contact with both K and Li metal[31,33]. The KFSI:DME electrolyte system has been shown to enable reversible plating and stripping of K metal due to formation of a more uniform solid electrolyte interphase (SEI) at the negative electrode, indicating an appropriate model electrolyte for the electrochemical measurements which require symmetric metal cells[31]. Moreover, Le Pham et al. have investigated the solvation structure of K⁺ in KFSI:DME electrolytes using operando XRD and Raman spectroscopy[34].

Our K metal preparation protocol enables improved K metal stability, opening up the potential for more accurate K-ion electrode and electrolyte characterisation[3,30]. By providing a comprehensive understanding of K-ion electrolyte transport, our work lays the foundation for more accurate Doyle-Fuller-Newman (DFN) modelling of KIBs and, most importantly, facilitates the development of high performance K-ion electrolytes.

## Results

### Potassium metal electrode preparation

State-of-the-art electrochemical techniques used to characterise electrolyte transport and thermodynamic properties rely on metallic electrodes that are sufficiently chemically and electrochemically stable in the electrolyte under investigation. The reactivity of metallic K is one of the reasons why a thorough characterisation of a K-ion electrolyte has not yet been reported. In this study, we developed a K electrode preparation protocol to ensure sufficient K metal stability and data reproducibility. The first step involves melting K metal chunks in an Ar-filled glovebox (<0.1 ppm O₂ and H₂O), skimming off impurities floating on the melt and quenching the clean metal. Right before cell assembly, a clean K metal sphere is rolled into a metal sheet (thickness ~ 0.6 mm), punched into discs and the surface polished using a microtoming technique adapted from a methodology developed for metallic lithium (Fig. 1)[35].

It has been shown that standard K metal preparation, where K metal is cut, washed in hexane, rolled, and then punched into electrodes[30,36] (Methods), can result in high open-circuit voltages (OCVs) in symmetric K cells, >100 mV, indicating K metal instability and inhomogeneous distribution of impurities[30,37]. Figure 2a shows the low OCVs and high precision results (Supplementary Note 1) for K symmetric cells prepared using our K electrode preparation protocol compared to the standard preparation procedure reported in literature[30,36], demonstrating improved surface stability and homogeneity[37]. This is also supported by the considerably reduced total impedance of cells prepared using our preparation compared to the standard method (Fig. 2b). The atomic force microscopy (AFM) height map in the inset in Fig. 2b further highlights the surface uniformity we are able to achieve with our preparation protocol, showing that the as-prepared K surface is flat with no visible contamination. The standard preparation method results in a surface with a much higher roughness (root mean square roughness of 253 and 33 nm for the

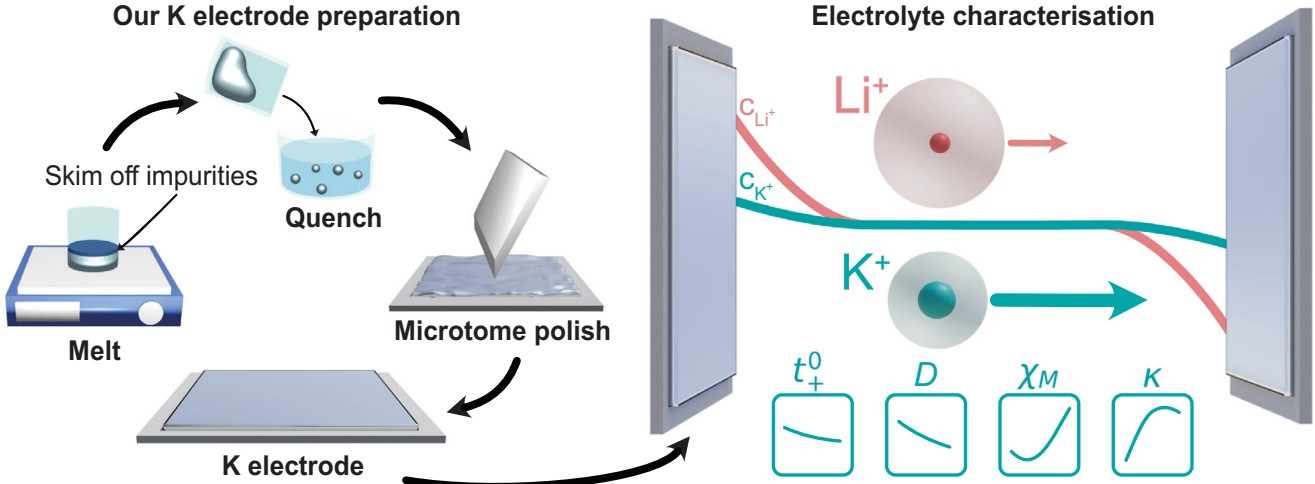

**Fig. 1 | Potassium electrode preparation and electrolyte characterisation.** Schematic of our K preparation protocol enabling K-ion electrolyte transport and thermodynamic property characterisation. Our K metal preparation involves melting K metal chunks, skimming off impurities floating on the melt and quenching the clean metal. Clean K metal spheres are rolled, punched into discs and the surface polished using a microtoming technique adapted from a

methodology developed for Li[35]. Our K preparation enables characterisation of the K-ion electrolyte salt diffusion coefficient ($D$), the cation transference number ($t_+^0$), the ionic conductivity ($\kappa$), and the thermodynamic factor ($\chi_M$). The higher $D$ and $t_+^0$ of KFSI:DME compared to LiFSI:DME results in reduced concentration gradient formation, which is represented schematically.

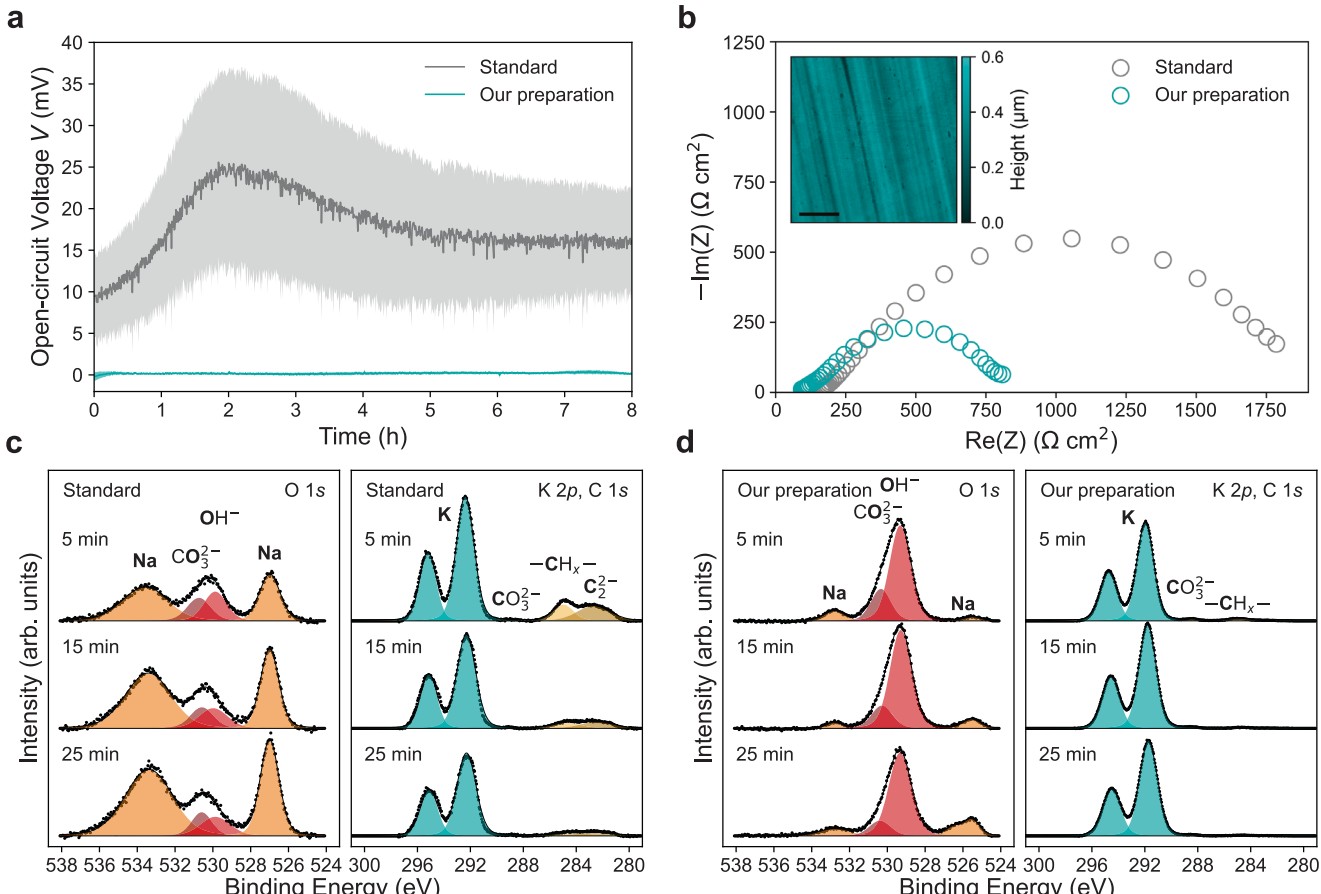

**Fig. 2 | Potassium electrode stability and characterisation.** Stability of K∥K symmetric cells in 1 m KFSI:DME using our K preparation compared to standard K preparation at 20 °C. **a** Averaged initial OCV profiles for 8 h. Light shaded areas depict standard error in the mean, calculated from at least 5 cells (Error analysis in Supplementary Note 1). **b** Complex impedance plots after 1 h rest at 20 °C, inset shows AFM height map of pristine K metal electrode prepared with our method (scale bar, 20 μm). XPS depth profiles on K metal after 5, 15 and 25 min of Ar⁺ sputtering (**c**) O 1s, K 2p and C 1s spectra from the standard preparation (**d**) O 1s, K 2p and C 1s spectra from our preparation.

surfaces prepared using the standard procedure and our preparation protocol, respectively) and greater nonuniformity, as evident in Supplementary Fig. 1. Crucially, our preparation protocol enables sufficient K stability for electrolyte transport property characterisation (Supplementary Note 2 and Supplementary Fig. 2). Whereas the standard preparation cannot be used to determine certain critical properties, such as $D$ (Supplementary Fig. 2).

X-ray photoelectron spectroscopy (XPS) with Ar⁺ depth profiling was utilised to examine the surfaces of K metal prepared using both the standard literature method and our preparation protocol to understand our improved stability (Supplementary Note 3). The O 1s spectra from the standard preparation in Fig. 2c exhibit peaks at 533.6 and 527.0 eV as a result of Na KLL Auger electron emission[38] and Na is the main species identified throughout the majority of the examined depth (Supplementary Figs. 3 and 4). However, Na is a minor impurity element in the electrode prepared with our method, as evidenced by the small Na KLL peaks in Fig. 2d (Supplementary Figs. 5 and 6). This Na-rich surface may impede the transport of K⁺ across the interface and be responsible for the large initial impedance observed (Fig. 2b). Additionally, inhomogeneous distributions of impurity elements like Na could alter the activity of the K metal electrodes, leading to the non-zero OCV evident in Fig. 2a[37].

The O 1s spectra from the electrode prepared with our preparation protocol are dominated by the KOH peak at 529.3 eV[38] in Fig. 2d. Due to the high reactivity of K metal this KOH likely forms within the XPS as fresh K metal is exposed, suggesting the surface is rich in

metallic K. In contrast, the hydroxide peak in Fig. 2c has a relatively low intensity and appears at a binding energy of 529.9 eV, consistent with NaOH[38]. This suggests there is limited metallic K available at the surface of the standard preparation electrode.

The K 2p XPS spectra from the standard K electrode in Fig. 2c show a decreasing K 2p doublet peak area with sputtering depth, while the K 2p doublet from our method remains intense during sputtering in Fig. 2d[39]. There is also a C 1s peak indicative of a carbide species at the surface of the standard K electrode in Fig. 2c (Supplementary Fig. 7 and Supplementary Note 3). These results therefore demonstrate that our K electrode preparation process produces K-rich electrode surfaces with greater uniformity and reduced levels of impurity elements, allowing K metal electrodes to be prepared with greater reproducibility.

K metal electrodes prepared with our protocol enable us to apply and adapt the most accurate electrochemical characterisation methods to measure the K-ion electrolyte $t_+^0$, $\kappa$, $\chi_M$ and $D$.

## Transference number

The cation transference number, $t_+^0$, is the fraction of current carried by the cation. It has been shown even increasing the transference number by 0.2 can significantly improve accessible capacity during non-aqueous Li-ion battery charge and discharge[40]. $t_+^0$ is difficult to measure accurately, and is often mischaracterised as the transport number using steady-state techniques such as the Bruce-Vincent method[41], relying on assumptions of electrolyte ideality and neglecting ionic species interaction[18,28]. Here the densitometric Hittorf method

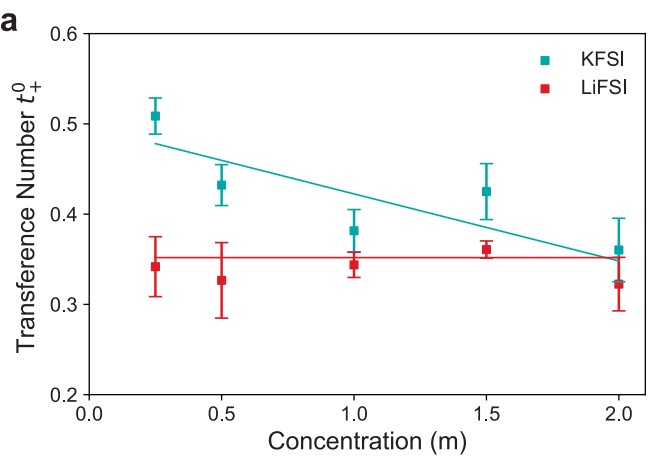

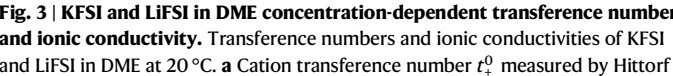

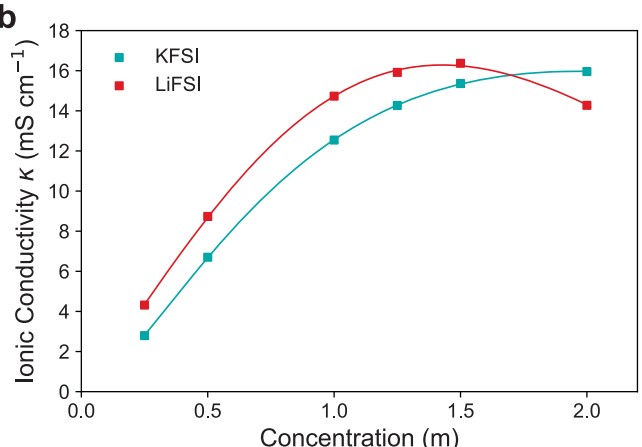

**Fig. 3 | KFSI and LiFSI in DME concentration-dependent transference number and ionic conductivity.** Transference numbers and ionic conductivities of KFSI and LiFSI in DME at 20 °C. **a** Cation transference number $t_+^0$ measured by Hittorf

experiments. Error bars for $t_+^0$ depict error in the mean (Supplementary Note 1). Fits described in Supplementary Note 9. **b** Ionic conductivity $\kappa$ measured with a conductivity cell, fit with the Casteel-Amis equation (Supplementary Eq. (6)).

was used to characterise the transference number relative to the solvent velocity[19,28,42,43]. This method involves applying a polarisation to a large symmetric cell, then closing two stopcocks to form three isolated chambers before extracting the solutions and measuring their densities to determine concentration changes (Methods, Supplementary Fig. 8 and Supplementary Note 4). The transference number is then determined via Eq. (1). The partial molar volume of salt, $\overline{V}_e$, of the K-ion and Li-ion electrolytes are used to determine $t_+^0$ (Supplementary Fig. 9 and Supplementary Note 5).

$$t_+^0 = 1 - \frac{FV_{chamber}|c_f - c|}{I_{pulse}t_{pulse}(1 - \overline{V}_e c)} \qquad (1)$$

Where $V_{chamber}$ is the volume of the cathodic or anodic cell chamber, $c_f$ is the concentration of the chamber after the experiment, $c$ is the concentration of the neutral chamber, $I_{pulse}$ is the current applied and $t_{pulse}$ is the pulse duration.

Figure 3a shows the valid transference numbers from the Hittorf measurements for KFSI and LiFSI in DME. The results show the $t_{K^+}^0$ is higher than $t_{Li^+}^0$ at lower concentrations of 0.25 m (0.49 and 0.34, respectively), though the $t_{K^+}^0$ decreases with increasing concentration so that it is is only slightly higher than $t_{Li^+}^0$ from 1.5–2 m. $t_{K^+}^0$ and $t_{Li^+}^0$ appear to be trending to similar values suggesting that the lower charge density of K$^+$ delays some of the ion-ion and ion-solvent interaction effects of increasing concentration. $t_{K^+}^0$ decreases from around 0.49 to 0.38 over the concentration range indicating increasing ion-ion and ion-solvent interactions are acting to bind up K$^+$ more strongly than FSI$^-$. Whereas for Li$^+$ these interactions appear to have become significant at concentrations below those measured as $t_{Li^+}^0$ remains constant in this concentration range. The $t_{K^+}^0$ results are similar to those reported for NaPF$_6$:EC:DEC by Landesfeind et al., though it is important to note their study treats the cosolvent as a single entity, assuming identical velocity of the two solvents, which is an assumption that has recently been shown by Wang et al. to have an impact on transference number measurements[43].

Supplementary Fig. 10 shows the anodic and cathodic chamber $t_+^0$ measurements, with $t_{K^+}^0$ exhibiting significant deviation between them. The cathodic data was discounted due to evidence of nonuniform K deposition and cathodic electrolyte discolouration in both systems (Supplementary Fig. 11). These high surface area K metal deposits result in the continuous formation of SEI, as indicated by the growing impedance in Supplementary Fig. 12, and significant electrolyte consumption. Therefore, given the highly sensitive density measurements, the cathodic transference numbers are likely

underestimated. There is evidence in the literature of nonuniform K deposition in the same electrolyte at much higher concentrations (5 M) and at lower current densities than can be used in our investigation (Supplementary Note 4)[31]. Significant discolouration was also observed in the cathodic solution for many of the Li- and K-ion experiments, further supporting the nonuniform Li/K deposition as mossy Li/K formed during plating can become electronically disconnected from the electrode, a phenomenon known as 'dead' metal formation[12]. Similar discolouration was found in the study by Hou and Monroe using metallic Li and their cathodic data was also discounted[19].

## Ionic conductivity

Figure 3b shows the ionic conductivities over the concentration range at 20 °C. LiFSI has higher $\kappa$ from low concentrations through until ~1.5 m, after which $\kappa$ decreases significantly, matching previous findings[44]. KFSI, however, continues increasing and appears to plateau at 2 m. It has been shown to decrease after this point[13]. This indicates there is likely significant species-species interaction for the LiFSI electrolyte compared with the KFSI electrolyte above ~1.5 m. The trend matches that found by Hosaka et al. for KPF$_6$ and LiPF$_6$ in EC:DMC where the K-ion electrolyte conductivity was lower than the Li-ion electrolyte below 1.5 m. However they also found KFSI:PC had significantly higher conductivity than LiFSI:PC at all concentrations tested[4]. This shows the importance of the combination of salt and solvent for optimum ionic conductivities for K-ion electrolytes. Since the $\chi_M$ for KFSI is closer to ideality than LiFSI at lower concentrations (Fig. 4), indicating less ion-ion interaction than for Li$^+$, it appears that the lower ionic conductivity for KFSI for the majority of the concentration range could be due to lower KFSI salt dissociation[4]. Plotting the equivalent conductance over concentration also indicates that both are weak electrolytes with their non-linear dependence of conductance on the square root of concentration[42,45] (Supplementary Fig. 13 and Supplementary Note 6). Though KFSI:DME is more non-linear than LiFSI:DME, again indicating lower salt dissociation for KFSI. At higher concentrations above 1.7 m, the drop in conductivity for LiFSI is likely due to the greater ion-solvent and ion-ion interaction for Li$^+$, where it is dragging more solvent than the K$^+$ due to its higher charge density and thus stronger solvation, while the stronger coulombic interaction of the Li$^+$ also results in greater ion-ion association, forming aggregates that increase the electrolyte viscosity[44,46]. Aggregates have been shown not to form in KFSI:DME until concentrations above this range (>3 M)[34]. High

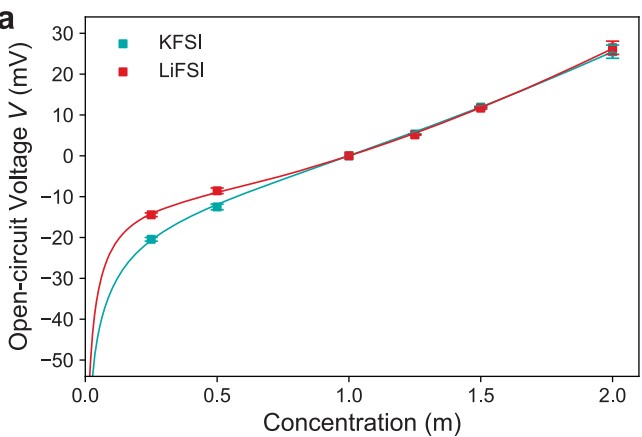
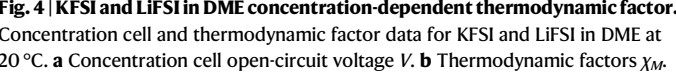
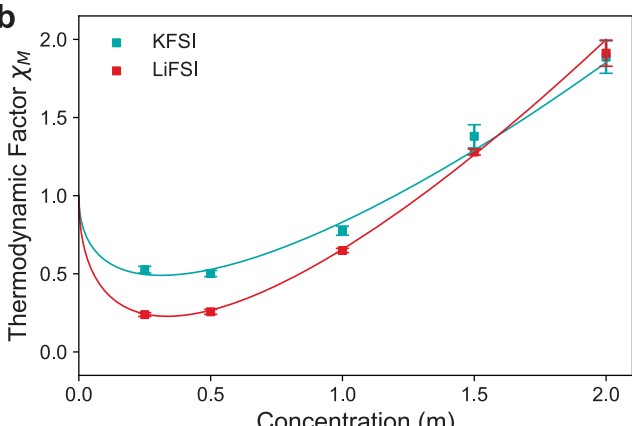

**Fig. 4 | KFSI and LiFSI in DME concentration-dependent thermodynamic factor.** Concentration cell and thermodynamic factor data for KFSI and LiFSI in DME at 20 °C. **a** Concentration cell open-circuit voltage $V$. **b** Thermodynamic factors $\chi_M$. Error bars depict the standard error in the mean for $V$, and the propagated $t_+^0$ error for $\chi_M$ (Supplementary Note 1). The fits are described in Supplementary Note 7.

ionic conductivities are reached in the DME electrolytes, 16 mS cm⁻¹ for KFSI:DME at 2 m and 15 mS cm⁻¹ for LiFSI:DME at 1.5 m. The $\kappa$ reached here are higher than that for carbonate equivalents due to the lower viscosity of DME[47], matching previous findings[13,48].

Supplementary Figs. 14 and 15 show that the activation energies of ionic conduction of KFSI and LiFSI in DME are very similar and increase with concentration as a result of increasing ion-solvent interactions[49]. The activation energy for both KFSI and LiFSI appears to be limited by the bulky FSI⁻ anion.

**Thermodynamic factor**

The thermodynamic factor, $\chi_M$, measures the non-ideality of an electrolyte and accounts for deviations from Nernstian behaviour, reflecting how the salt thermodynamic activity varies with concentration. Concentration cells were used in the measurement of $\chi_M$ where the open-circuit voltages were measured between 'test' and 'reference' solutions (Methods and Supplementary Fig. 16). The change in the OCV across the concentration cell, $V$, with molar concentration, $c$, is related to the thermodynamic factor, $\chi_M$, and transference number, $t_+^0$, by Eq. (2)[18]:

$$\chi_M = 1 + \frac{d\ln(f_{\pm})}{d\ln(c)} = \frac{F}{2RT(1-t_+^0)}\frac{dV}{d\ln(c)} \quad (2)$$

Where $f_{\pm}$ is the mean molar activity coefficient, $F$ is the Faraday constant, $R$ is the gas constant and $T$ is the absolute temperature. The solvent concentration, $c_0$, and the partial molar volume of solvent, $\overline{V}_0$, can be used to map the thermodynamic factor to the molar basis from the molal basis in which it is defined[50,51]. As derived in Supplementary Note 7 (Supplementary Figs. 17 and 18), we fit $\chi_M$ to the function given in Eq. (3):

$$\chi_M = \frac{1}{c_0\overline{V}_0}\left(1 + A_1 c_m^{1/2} + A_2 c_m\right) \quad (3)$$

Where $c_m$ is the molal concentration and $A_1$ and $A_2$ are fitting constants. The fits to the OCV data are presented in Fig. 4a.

Figure 4b shows how the thermodynamic factor changes with concentration. The trends match those found for other non-aqueous electrolytes[18,19,27,42]. With increasing concentration, first coulombic ion-ion interactions decrease the salt free energy relative to the DME, reducing the salt activity coefficient and hence causing a drop in $\chi_M$. As the concentration increases further, ion-solvent interactions increase, resulting in DME being increasingly bound, decreasing solvent vapour

pressure, hence increasing the salt activity coefficient and $\chi_M$[42,52]. The results show the decrease in $\chi_M$ at lower concentrations for KFSI is significantly less than for LiFSI. This can be attributed to the following two factors. First the larger size and thus lower charge density of K⁺ compared with Li⁺, resulting in weaker K⁺ coulombic ion-ion interactions, hence the smaller reduction in $\chi_M$. Second, from Debye-Hückel theory the gradient of the $\chi_M$ decrease at lower concentrations for both LiFSI and KFSI in DME should be equal as it is determined by the dielectric constant, $\varepsilon$, of the solvent[51]. As Debye-Hückel theory assumes fully dissociated electrolytes, the lower KFSI gradient magnitude could further suggest poorer salt dissociation[42], which is also indicated by the lower ionic conductivity (Fig. 3b) and more non-linear equivalent conductance (Supplementary Fig. 13) at low concentrations. The low $\varepsilon$ of DME also provides minimal electrostatic screening, thus having a limited effect on reducing ion-ion interactions[53].

At higher concentrations, the increase in $\chi_M$ with increasing concentration is also higher for LiFSI than for KFSI. This is again due to the reduced charge density of K⁺ resulting in weaker interactions with solvent molecules and thus a smaller solvation shell of K⁺ compared to Li⁺. The weaker K⁺ solvation results in more free DME compared to Li⁺ at higher concentrations, and therefore, the salt activity coefficient and $\chi_M$ increase with increasing salt concentration at a reduced rate for KFSI. This is the same trend identified by Landesfeind et al. comparing NaPF₆ and LiPF₆ in EC:DMC[16]. Le Pham et al. found a constant solvation number for KFSI:DME from Raman characterisation across this concentration range, supporting this increased binding of solvent increasing $\chi_M$[34]. A similar study also found significant ion-pairing and solvent binding in LiFSI:DME[44]. $\chi_M$ is below unity until relatively high concentrations for both LiFSI and KFSI (~1.4 m) indicating the point where ion-ion and ion-solvent interactions are equal.

Further experiments could characterise $\chi_M$ over a wider concentration range with greater accuracy using the shifting-reference concentration cell technique developed by Wang et al.[42]. Given the narrow concentration range, this was not deemed necessary for this study.

**Diffusion coefficient**

Steady-state polarisation and long-term relaxation restricted diffusion[19,54–57] was used to characterise $D$ as it has been identified as being more accurate than pulse polarisation methods, being less susceptible to double layer relaxation effects[57]. A custom restricted diffusion cell was designed (Supplementary Fig. 19) and galvanostatic polarisation was used to form the concentration gradient which was subsequently relaxed (Methods and Supplementary Note 8). Contrary

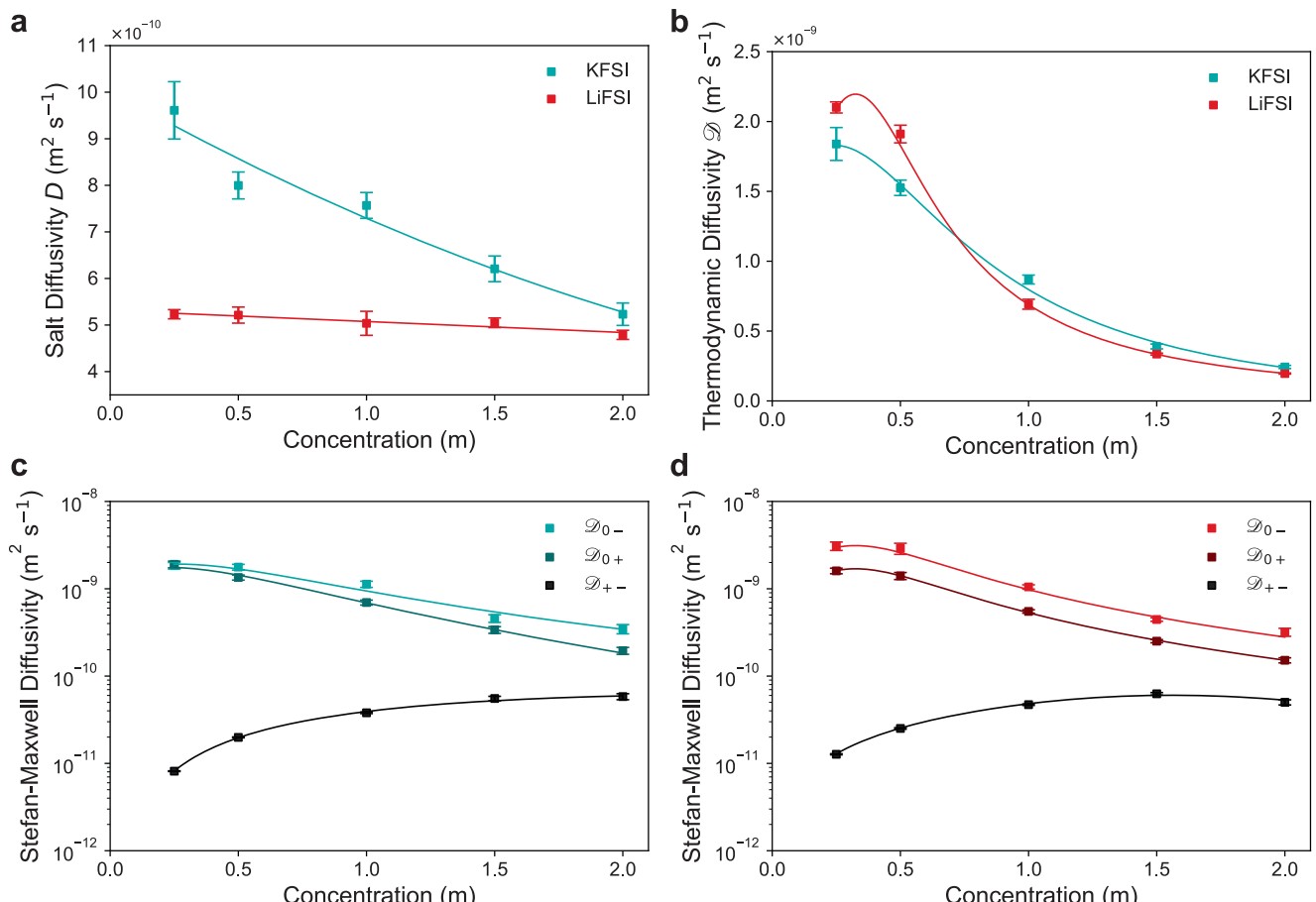

**Fig. 5 | KFSI and LiFSI in DME concentration-dependent diffusivities.** Diffusion coefficients of KFSI and LiFSI in DME at 20 °C measured by steady-state galvanostatic restricted diffusion. **a** Salt diffusion coefficient $D$. Error bars depict the standard error in the mean (Supplementary Note 1). Fits described in Supplementary Note 9. **b** Thermodynamic diffusion coefficient $\mathscr{D}$ calculated using the measured thermodynamic factor (Fig. 4b). Stefan-Maxwell diffusion coefficients $\mathscr{D}_{ij}$ for **c** KFSI and **d** LiFSI. Errors bars in $\mathscr{D}$ and $\mathscr{D}_{ij}$ depict the propagated $D$, $t_+^0$ and $\chi_M$ errors (Supplementary Note 1). Fits obtained by combining all parameterised transport and thermodynamic properties (Supplementary Note 9).

to many recent studies[16,19,57], no separators were used in the restricted diffusion experiments to improve accuracy, due to the errors introduced through separator tortuosity estimation and variability[19,42]. The cell was oriented vertically to suppress natural convection[19,51]. Exponential relaxation occurs where the concentration gradient relates to the OCV, and the diffusion coefficient, $D$, can be determined from the linear time dependence of the logarithm of the open-circuit voltage via Eq. (4) (Supplementary Fig. 20). The cell geometry was designed to enable longer polarisation and relaxation times (20 h for polarisation and up to a maximum of 60 h for relaxation) to ensure less noisy relaxation and a more robust fit for both KFSI and LiFSI, similar to Wang et al.[28]. This was significantly longer than the shorter relaxation times (≤3 h) in many recent studies[16,20,57].

$$\frac{d\ln(V)}{dt} = -\frac{\pi^2 D}{L_s^2} = -\frac{1}{\tau_{diff}} \quad (4)$$

where $V$ is the restricted diffusion cell OCV measured during relaxation, $L_s$ is the bulk electrolyte thickness, $t$ is time and $\tau_{diff}$ is the characteristic decay time $= \frac{L_s^2}{\pi^2 D}$.

Figure 5a shows $D$ is higher for KFSI than for LiFSI at all concentrations at 20 °C (Supplementary Fig. 21 shows all data). At 1 m $D_{KFSI}$ is over 50% higher than $D_{LiFSI}$ (7.6 × 10$^{-10}$ m² s⁻¹ and 5.0 × 10$^{-10}$ m² s⁻¹, respectively). The difference is most significant at low concentrations where $D_{KFSI}$ is almost double that for $D_{LiFSI}$ (9.6 × 10$^{-10}$ m² s⁻¹ and 5.2 × 10$^{-10}$ m² s⁻¹, respectively at 0.25 m). Supplementary Fig. 22 shows

the faster relaxation profile and time of KFSI compared to LiFSI at 0.25 m. With increasing concentration the difference between $D_{KFSI}$ and $D_{LiFSI}$ becomes smaller (5.2 × 10$^{-10}$ m² s⁻¹ and 4.8 × 10$^{-10}$ m² s⁻¹, respectively at 2 m). This appears again to be due to increasing ion-ion and ion-solvent interactions occurring for K⁺ at higher concentrations having a greater relative impact compared to minimal interaction at lower concentrations, matching the same trend observed for $t_+^0$. Both $D_{KFSI}$ and $D_{LiFSI}$ appear to be trending to similar values with increasing concentration, again matching the trends for $t_+^0$ and supporting the argument that the lower charge density appears to delay some of the ion-ion and ion-solvent interaction effects of increasing concentration. The higher $D_{KFSI}$ also matches the trend observed by Landesfeind et al. with NaPF$_6$ showing higher $D$ than LiPF$_6$[16]. However, the difference is much more significant for the K-ion electrolyte. $D_{KFSI}$ is also significantly higher than those of Li-ion and Na-ion electrolytes characterised, with 2.9 × 10$^{-10}$ m² s⁻¹ found for 1 M NaPF$_6$:EC:DEC at higher temperature of 25 °C[16], and with $D$ for the majority of LiPF$_6$-based electrolytes coalescing around 2–3 × 10$^{-10}$ m² s⁻¹ at 1 M at 25 °C[18,19,42,57]. The values obtained for $D_{LiFSI}$ here are also higher than most Li-ion carbonate electrolytes characterised and this is attributed to the significantly lower viscosity of the DME solvent used compared to carbonate solvents[53].

Figure 5b shows $D$ converted into the thermodynamic diffusion coefficient, $\mathscr{D}$, using $\chi_M$ and Supplementary Eq. (17), reflecting the diffusion coefficient with respect to salt chemical potential gradients instead of concentration gradients. The $\mathscr{D}$ trend matches those in literature[19,22,27,42]. The initial increase in $\mathscr{D}$ is due to increasing ion

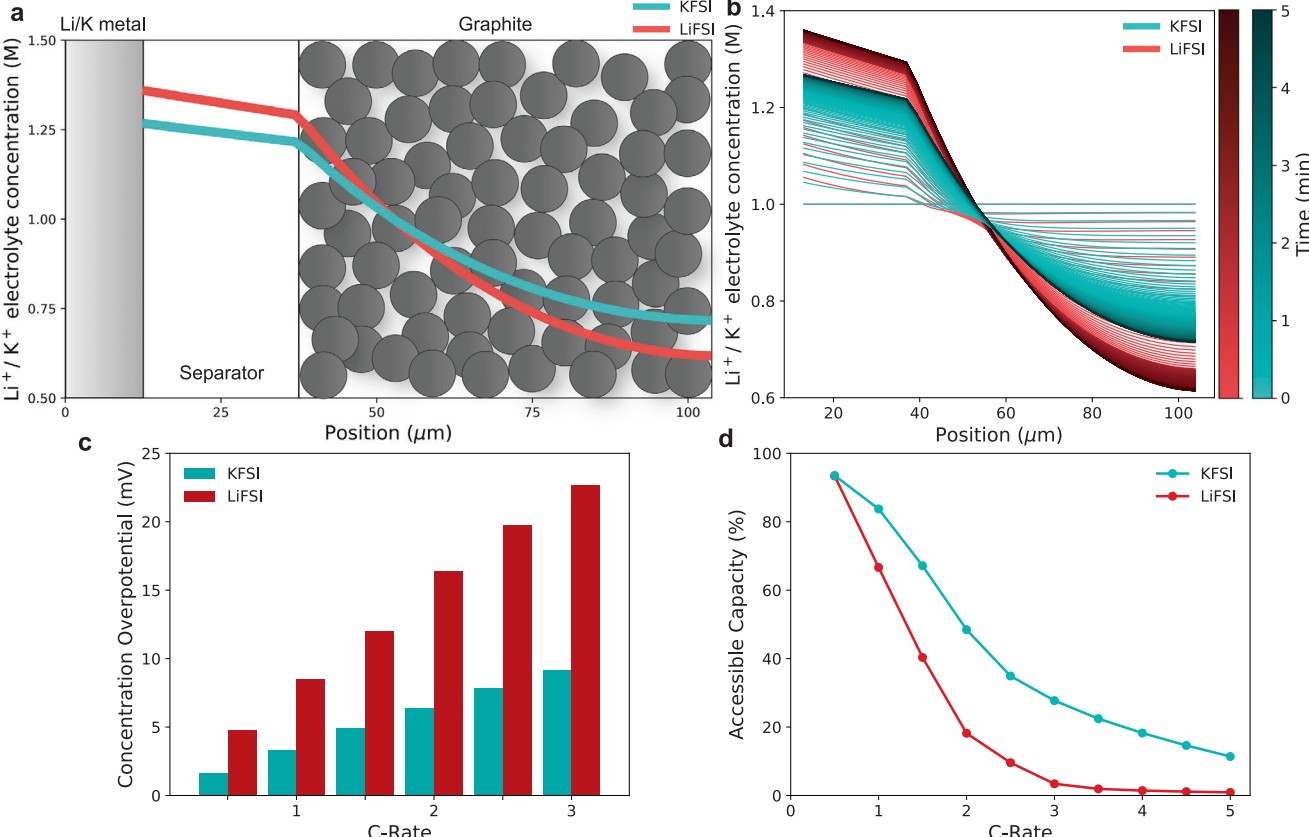

**Fig. 6 | Doyle-Fuller-Newman (DFN) simulations.** DFN simulations of K‖graphite and Li‖graphite cells during charge with KFSI and LiFSI in DME electrolytes using the transport and thermodynamic property relationships characterised (Supplementary Note 9). Modelled using PyBaMM[59]. **a** Simulated metal‖graphite cells after 5 min of 2C constant current charge (1C = 3.28 mA cm⁻² for both) including Li⁺/K⁺ electrolyte concentration gradient. **b** Li⁺/K⁺ electrolyte concentration gradient over time during 2C charge. **c** Concentration overpotentials for the metal‖graphite cells with increasing C-rate. **d** Accessible capacity for the metal‖graphite cells with increasing C-rate.

association due to the effective merging of two species into a single species resulting in lower resistance to the single species motion[42]. $\mathscr{D}_{LiFSI}$ shows a greater initial increase due to greater ion-ion interaction for Li⁺ than for K⁺ as indicated by $\chi_M$. At higher concentrations $\mathscr{D}_{KFSI}$ is only slightly higher than $\mathscr{D}_{LiFSI}$, but shows almost the same trend, demonstrating that the most significant difference in diffusion behaviour is related to concentration gradients rather than chemical potential gradients.

The Stefan-Maxwell diffusion coefficients express the mobility of each electrolyte species relative to each other in terms of the thermodynamic forces driving diffusion, providing deeper understanding of diffusional behaviour (Supplementary Note 8). The Stefan-Maxwell coefficients for the KFSI and LiFSI electrolytes are shown in Fig. 5c, d, respectively. The coefficients are relatively similar for both KFSI and LiFSI and this is due to their similar thermodynamic diffusivities. Both of the solvent-ion coefficients, $\mathscr{D}_{0+}$ and $\mathscr{D}_{0-}$, for LiFSI and KFSI decrease by about an order of magnitude over the concentration range, demonstrating the drag from the DME solvent becomes stronger on both cation and anion with increasing salt concentration. $\mathscr{D}_{0-}$ is higher than $\mathscr{D}_{0+}$ across the concentration range for LiFSI, indicating weaker interaction of the FSI⁻ with the DME compared with the Li⁺ interaction with DME. However, for KFSI, $\mathscr{D}_{0-}$ and $\mathscr{D}_{0+}$ are much closer and at low concentrations are almost the same. This is due to $t_+^0$ being ~ 0.5 for K⁺ at 0.25 m indicating K⁺ and FSI⁻ are carrying the same amount of current. With increasing concentration, the difference between $\mathscr{D}_{0-}$ and $\mathscr{D}_{0+}$ increases, matching the falling $t_{K^+}^0$ as ion-solvent interactions increase. The ion-ion diffusivity $\mathscr{D}_{+-}$ for both LiFSI and KFSI is approximately two orders of magnitude lower than the

solvent-ion diffusivities, particularly at lower concentrations demonstrating ion-ion interaction is somewhat significant for both, similar to that found for LiPF₆:PC[19], but not as significant as the five orders of magnitude difference found for LiPF₆:EMC, indicating substantial ion-ion interaction[42]. For both KFSI and LiFSI the maximum in $\mathscr{D}_{+-}$ matches their maximum ionic conductivity, suggesting the greater cation/anion interaction is occurring due to lack of free DME at a lower concentration for Li⁺, corresponding to its stronger solvation interactions.

## Modelling

To understand the impact of the differences in the electrolyte transport and thermodynamic properties on cell performance, we conducted Doyle-Fuller-Newman (DFN)[18,58] simulations of the charging behaviour of K-ion and Li-ion graphite half-cells with KFSI and LiFSI:DME electrolytes. The four properties characterised here ($D$, $t_+^0$, $\kappa$ and $\chi_M$) are all of the electrolyte properties required for DFN cell modelling, therefore, this is the first time K-ion has been simulated using the DFN model. The cells were modelled using the battery modelling package PyBaMM[59] in combination with the empirical transport and thermodynamic property relationships of the KFSI and LiFSI:DME electrolytes characterised in this paper (Supplementary Note 9). Since graphite is the negative electrode of choice for both Li-ion and K-ion[3,5,60] and is the limiting factor in realising fast charging in Li-ion batteries[60,61], the metal cell using graphite as the working electrode was deemed the most appropriate comparison. Full details of the model are described in the Methods.

Figure 6a depicts the metal‖graphite cell being modelled and shows the reduced electrolyte concentration gradient formation for

the KFSI cell compared to the LiFSI cell after 5 minutes of charging at 2C (1C = 3.28 mA cm$^{-2}$ for both). The LiFSI cell reaches the lower cut-off voltage of 0.01 V very shortly after this due to overpotentials, below which Li metal plating would start to occur[61]. Figure 6b shows the concentration gradient formation over time at the same charging rate, again showing reduced electrolyte concentration gradient formation for the KFSI cell and demonstrating the clear advantages of the higher $t_+^0$ and $D$ for the KFSI electrolyte. Figure 6c shows that the reduced concentration gradient formation results in lower concentration overpotentials ( ~ 60% lower for the KFSI compared to the LiFSI cell at the charging rates 1–3C) again emphasising the importance of $t_+^0$ and $D$. Finally, Fig. 6d illustrates the impact on fast charging cell performance, demonstrating that the KFSI cell can achieve higher accessible capacities at higher charging rates (48% vs. 18% at 2C for the KFSI cell compared to the LiFSI cell, respectively), showing improved high-power capability for the KFSI cell. The greater Li-ion concentration overpotentials cause the LiFSI cell to reach the lower cut-off voltage, and hence the Li plating potential, faster than for the KFSI cell, limiting the accessible capacity. From the reduced K$^+$ electrolyte concentration gradient formation in Fig. 6a, b, the electrolyte K$^+$ concentration at the back of the graphite electrode near the current collector is considerably higher than that for Li$^+$, enabling greater accessible K$^+$ for graphite intercalation and hence greater accessible capacity. Even at the high charging rate of 4C the KFSI cell can access 19% capacity whilst the LiFSI cell only 1%. These results demonstrate the important role of the electrolyte transport properties in increasing high-power performance, particularly $t_+^0$ and $D$. The faster electrolyte transport properties combined with the higher potential of K$^+$ intercalation into graphite[3,10], enables improved high-power charging performance of the K-ion chemistry compared to Li-ion.

Nevertheless, this is only a provisional indicative model to demonstrate the potential impact on cell performance of the faster K-ion electrolyte transport properties. Full-cell high-power K-ion performance may only be achieved if a suitable electrolyte is developed which provides both a stable SEI for the graphite anode and stability at the high operating voltages of the leading cathodes[3]. Also for a more accurate and sophisticated full-cell model, more advanced characterisation should be conducted of critical electrode properties such as the solid diffusivities and exchange current densities.

## Discussion

In summary, we have fully characterised the ionic transport and thermodynamic properties of a K-ion electrolyte system and compared them to the Li-ion equivalent. We developed a K metal preparation protocol which enabled sufficient stability for electrolyte characterisation. The results show the salt diffusion coefficient and cation transference number of the KFSI:DME electrolyte are significantly higher than that of the LiFSI electrolyte for all concentrations below 2 m. Higher salt diffusion coefficients and cation transference numbers reduce ionic concentration gradient formation and the associated concentration overpotentials, thus substantiating the potential of KIBs to deliver improved rate capability and low-temperature performance. The ionic conductivities were found to be similar at 20 °C, with LiFSI slightly higher until ~ 1.7 m, likely due to inadequate KFSI salt dissociation. The thermodynamic factor behaviour with concentration appears to indicate weaker solvent and ion-ion interactions of K$^+$ compared to Li$^+$. DFN simulations of K-ion and Li-ion metal‖graphite cells, using the electrolyte property relationships characterised here, demonstrates the faster transport properties of the KFSI:DME electrolyte results in improved charging rates. Overall this study proves that the increased cation size and lower charge density of K$^+$, and thus weaker solvent and ion-ion interactions are beneficial for high-power electrochemical energy storage systems. Full characterisation of the K-ion electrolyte has provided a more accurate understanding of K-ion electrolyte mass transport and thermodynamics, laying the foundations for further K-ion electrolyte development and optimisation.

## Methods

### Electrolyte preparation and electrochemical measurements

All electrolytes were prepared and handled in an Ar-filled glovebox with O$_2$ and H$_2$O concentrations below 0.1 ppm. The electrolyte used was a solution of potassium bis(fluorosulfonyl)imide (KFSI, 99.9% Solvionic) or lithium bis(fluorosulfonyl)imide (LiFSI, battery grade, Fluorochem) in 1,2-dimethoxyethane (DME, 99.5% anhydrous, Sigma Aldrich). KFSI was dried under a high vacuum at 100 °C for 48 hours and LiFSI at 70 °C for 48 h. DME was dried using 3 Å molecular sieves. All equipment was dried at 70 °C under vacuum for a minimum of 24 hours before being used and brought into the glovebox. The H$_2$O content of the electrolyte solutions was determined by Karl Fischer titration, also performed in an argon-filled glovebox, and recorded to be below 10 ppm of H$_2$O. All restricted diffusion, Hittorf and concentration cell experiments were conducted in a Binder Oven at 20 °C ( ± 0.3 K).

All electrochemical tests were carried out using a battery cycler (VMP3, Biologic). Electrochemical impedance spectroscopy (EIS) measurements were performed using a frequency response analyser (VMP3, Biologic), unless otherwise stated, over the frequency range of 200 kHz–500 mHz (6 measurement points per decade) with an applied potentiostatic signal of amplitude 10 mV. The spectra in Fig. 2b were gathered at OCV after a 1 h rest. The spectra in Supplementary Fig. 12 were gathered every 30 min during OCV relaxation after a 20 h current pulse.

### Electrode preparation

For the potassium electrode preparation protocol, potassium electrodes were prepared from potassium chunks in mineral oil (98% trace metals basis, Sigma Aldrich). First the K chunks were removed and melted in a beaker on a hot plate in an argon-filled glovebox (<0.1 ppm O$_2$ and H$_2$O). A spatula was then used to skim off and remove the visible impurity layers until the liquid K metal appeared clean. Then the liquid K metal was quenched into clean mineral oil forming spheres of clean K. These K spheres were then cleaned with hexane (95% anhydrous, Sigma Aldrich). Just before use the K was rolled to ~ 0.6 mm thickness using an aluminium rolling pin with the K sandwiched between two sheets of weighing paper (Grade 2122, Whatman) coated in hexane and one metal surface was gently polished using a plastic blade to remove any oxide and provide a sticking surface. Electrodes were then punched into discs of required diameter using a wad punch. The K electrode was placed on the current collector (stainless steel) with the polished surface down. Next, for the active and exposed K surface, first the K was initially polished with the plastic blade, then followed by a second careful polish using a microtome blade (polytetrafluoroethylene coated, Epredia, Shandon), adapting a methodology developed for metallic lithium[35]. The microtome blade was used to form a mirror-like finish, resulting in an improved polished K surface free of surface irregularities. The active K metal surface was polished at the very end of cell setup, just before electrolyte addition, so the polished surface was exposed to the glovebox environment for minimal time before the electrolyte was added.

For the standard preparation the same method was used from literature[30,36]. K metal was cut, washed in hexane, and rolled before punching into electrodes. Lithium (99.9% trace metal basis, Sigma Aldrich) electrodes were prepared for use by first initially brushing the Li metal surfaces using a plastic brush, then calendering the brushed Li to 0.3 mm thickness in a sample bag, before finally punching into electrode discs of required diameter using a wad punch. Stainless steel current collectors were also used for the Li cells.

## Atomic force microscopy

Atomic force microscopy (AFM) was performed with a Bruker Dimension Icon AFM in an argon-filled glovebox (<0.1 ppm $O_2$ and $H_2O$). Surface height maps were gathered in the ScanAsyst imaging mode with ScanAsyst-Air probes (Bruker) at a scan rate of 0.25 Hz. Gwyddion software was used for data analysis[62].

## X-ray photoelectron spectroscopy

X-ray photoelectron spectroscopy (XPS) was performed with an ULVAC PHI Versaprobe III XPS system generating monochromatic Al$_{K\alpha}$ X-rays (1486.6 eV, 15 kV anode voltage, 25 W beam power) under ultrahigh vacuum (UHV) conditions ($\sim 10^{-7}$–$10^{-6}$ Pa). K metal samples were prepared in a glovebox and were immediately transferred into the XPS chamber using a vacuum transfer vessel (ULVAC PHI GmbH) to avoid contamination and ambient exposure. A 500 μm × 500 μm area from each sample was analysed. Survey scans were acquired at pass energies of 224 eV, and a lower pass energy of 55 eV was used for core-level spectra. In-built electron and low energy Ar$^+$ sources were utilised for charge neutralisation. Depth-profiling was achieved with consecutive XPS analysis and Ar$^+$ sputtering (4 keV, 3 mm × 3 mm) for a total of 60 min. Acquired spectra were fitted with Voigt lineshapes, after application of a Shirley background, using CasaXPS software[63]. All spectra were charge referenced to adventitious C 1$s$ peak at 285 eV[38]. Fitted regions were quantified and relative fractions of components were estimated using the relative sensitivity factors (RSFs) provided by CasaXPS (Supplementary Figs. 3 and 5)[63].

## Densitometry

For greater accuracy electrolyte concentrations were prepared gravimetrically rather than volumetrically, as using an analytical balance is more precise than a volumetric flask. In order to convert the gravimetric concentrations to volumetric (molality to molarity), high precision 5-digit density measurements were obtained using an Anton Paar DMA 4100 density meter in an argon-filled glovebox (<0.1 ppm $O_2$ and $H_2O$). Each measurement was temperature controlled at 20 °C. The density meter was rinsed with isopropanol (≥99.9%, HPLC grade, Fisher Chemical) and DME (99.5% anhydrous, Sigma Aldrich) at least three times and dried in ambient argon between measurements. It was ensured that the density meter was completely clean and returned to reading the argon density between each measurement. The density correlations for KFSI and LiFSI in DME are shown in Supplementary Fig. 8.

## Hittorf method

The sealed Hittorf cell was oriented vertically in the Binder Oven, and the current was applied so stripping occurred at the bottom electrode (anodic) and plating at the top electrode (cathodic), to prevent natural convection effects[19,51]. After an initial rest of 4 h, the current polarisation was applied for duration $t_{pulse}$ = 20 h with the stopcocks open where the cell consists of a single cavity. Once finished, the two stopcocks were immediately closed creating three isolated chambers: anodic chamber at the bottom where stripping occurred, neutral chamber in the middle, and cathodic chamber at the top. The electrolyte solutions were then extracted through access ports. Extracted solutions from the three chambers were stirred for at least 1 h to ensure uniform concentration, after which their densities were measured using the Anton Paar DMA 4100 density meter at 20 °C. The molarity of the solutions was calculated using the density correlation (Supplementary Fig. 8). The differences in concentrations of the anodic and cathodic chambers from the neutral chamber was used to calculate $t_+^0$. The current used for polarisation for all K-ion Hittorf experiments was 100 μA, except at 0.25 m where 50–100 μA was used. For the Li-ion Hittorf experiments 200 μA was used except at 0.25 m where 50 μA was used. $I_{pulse}$ and $t_{pulse}$ were set such that the concentration boundary layers remained within the anodic and cathodic chambers during the experiment[64]. Three measurements were taken at each concentration for KFSI and LiFSI concentrations above 0.5 m. Two measurements were taken for LiFSI at 0.25 m and 0.5 m.

## Ionic conductivity

For measurement of ionic conductivity, a commercial conductivity cell of known cell constant was used (CLR, 401-S-138C). The cell was filled with ~0.5 mL electrolyte and tested with an impedance analyser (Bio-Logic MTZ-35 with ITS-e temperature chamber) at 15, 20 and 25 °C. The ionic conductivity of the electrolyte was calculated by dividing the cell constant by the series resistance extracted from a Nyquist plot. The ionic conductivities in Fig. 3b were fitted with the function proposed by Casteel and Amis (Supplementary Eq. (6))[65].

## Concentration cell

For the concentration cell experiments a H-cell was designed including a Grade 5 frit to mitigate the faster diffusion from the K-ion electrolyte. A Grade 4 frit was found to be insufficient to suppress interdiffusion in the K-ion electrolyte, but gave reliable results with the Li-ion electrolyte. K and Li metal were prepared and cut into ~5 mm × 20 mm strips. Each chamber of the H-cell was filled with 4 mL of electrolyte, with 1 m electrolyte used as the constant 'reference' concentration for both electrolyte systems. The electrodes were then lowered into the electrolyte and the cell was sealed and immediately brought into the Binder Oven. The OCV was tracked for 2 h to allow the cell to stabilise and reach the correct temperature and the OCV was then averaged over the next 10 min, as shown in Supplementary Fig. 16. At least three repeat measurements were made for each concentration.

## Restricted diffusion

The cell used for the restricted diffusion cells (Supplementary Fig. 19) was designed to ensure an airtight seal and the chamber is free of any geometric issues that can affect the concentration gradient. The cell was designed to be longer than typically used for Li-ion restricted diffusion experiments due to the identified faster diffusion of K-ion electrolytes to enable sufficient time to observe the relaxation and enable a more robust fit. The distance between the current collectors was 12 mm and the distance between the electrodes, $L_s$, was measured using digital calipers for each cell due to the slightly varying thickness of K. The thickness of the K was measured once it had been placed on the cell current collector due to K being soft and easily compressed. For example, $L_s$ was 10.8 mm with K metal electrodes of 0.6 mm thickness. No separators were used to improve accuracy errors introduced through separator variability and tortuosity estimation[19,42].

The experiment involved first a rest for 10 h where the OCV was tracked. Then a galvanostatic polarisation was applied for 20 h to induce the concentration gradient. The current was 35 μA for all Li-ion cells and 30 μA for K-ion cells from 1 m and higher concentration. 25 μA current was used for 0.25 and 0.5 m for K-ion. Finally, the current was switched off and the OCV recorded during relaxation. At least five cells were made for each K-ion concentration and at least three for Li-ion (Supplementary Fig. 22). The OCV values were adjusted by any $V_{offset}$ from 0 V to ensure the OCV relaxed to 0 V so the linear behaviour of $\ln(V)$ vs time could be analysed[16,57]. The data was fit from the minimum time constant 0.5 $\tau_{diff}$ established by Newman and Thompson[66] for as long as it showed exponential relaxation behaviour, or until relaxation had completed (Supplementary Note 8). The $\ln(V)$ vs. time was plotted, with the gradient from the linear region used to obtain the salt diffusion coefficient. Representative relaxation profiles and the linear $\ln(V)$ vs. time for 1 m KFSI and LiFSI in DME are shown in Supplementary Fig. 20.

## Modelling

For the DFN modelling, the open-source battery simulation package Python Battery Mathematical Modelling (PyBaMM)[59] version 23.2

half-cell DFN model and CasADi numerical solver[67] was used. For determining the accessible capacity % at different C-rates, the baseline performance was determined from a C/50 charge rate (1C = 3.28 mA cm$^{-2}$ for both). For the KFSI and LiFSI:DME electrolytes the empirical concentration dependent functions characterised in this paper were used (Supplementary Note 9). The 'Ai2020' pouch cell parameter set was used for the Li graphite and cell geometry base parameters[68]. The default metal electrode parameters were used[69] with a metal electrode thickness of 12.5 μm. Between the Li-ion and K-ion graphite, electrode parameters such as particle size, porosity, and graphite electrode geometries were kept constant as defined in the parameter set[68]. The graphite diffusivity function was also kept constant since the K graphite diffusivities characterised so far are similar to that of Li[18,70,71]. The exchange current density function was also kept constant as no K$^+$ graphite exchange current density has yet been characterised. We incorporated the K graphite OCV profile for the K graphite simulation based on characterised data[72]. The lower voltage cut-off was set as 0.01 V vs. K$^+$/K or Li$^+$/Li below which metal plating would occur.

## Data availability

All the experimental data used in this study are available in the Zenodo database under a Creative Commons Attribution 4.0 International License (https://doi.org/10.5281/zenodo.8014257)[73].

## Code availability

The Python codes used in the DFN modelling and the diffusivity and thermodynamic factor analysis are available in the Zenodo database under a Creative Commons Attribution 4.0 International License (https://doi.org/10.5281/zenodo.8014257)[73].

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

## Acknowledgements

The authors would like to acknowledge the financial support of the Henry Royce Institute (through UK Engineering and Physical Sciences Research Council grant EP/R010145/1) for capital equipment. S.D. appreciates the financial support from EPSRC and Shell. B.J. is grateful for the support of the Clarendon Fund Scholarships. A.M. acknowledges the support of The Faraday Institution (via grant number FITG-FUSE-102). We are grateful to Christopher Doerrer for his help with the CAD for the final diffusion cell, Isaac Capone for his earlier teaching in preparing K, Giulia Galatolo and Soochan Kim for their help with the schematic design, Marco Siniscalchi for his earlier electrode preparation, Robert Timms for answering questions on PyBaMM, Maximilian Schart for his useful thoughts and ideas, Jack Fawdon for his earlier mentoring on transport, and Peter Klusener for his useful feedback.

## Author contributions

S.D. and M.P. conceptualised the study and experiments; S.D. developed the electrolyte characterisation cells and K preparation protocol;

S.D., B.J. and M.P. developed the potassium electrolyte characterisation methodology; S.D., B.J. and A.M. conducted the electrolyte characterisation experiments; B.J. conducted the XPS and AFM experiments; S.D. and B.J. conducted the analysis; S.D. conducted the modelling; S.D. wrote the original draft; S.D., B.J. and M.P. wrote, edited and revised the manuscript; M.P. supervised the study and provided frequent input in interpretation of results.

## Competing interests

M.P. is a scientific advisor to Project K Energy. The remaining authors declare no competing interests.
