## [Peer Review File · Nature Communications]

REVIEWER COMMENTS

Reviewer #1 (Remarks to the Author):

The authors have submitted the paper "Characterising the Ionic Transport and Thermodynamic Properties of Potassium-ion Electrolytes" which is a comparative study of the non-aqueous K-ion electrolyte KFSI:DME with the well-known Li-ion counterpart LiFSI:DME.

Indeed, the understanding of the ionic transport and thermodynamic properties of LiFSI:DME is crucial for the further development of K-ion batteries. The authors claim that "there is currently no study which has fully characterised the ionic transport and thermodynamic properties for nonaqueous K-ion electrolytes." Back in 2017, the long-term and highly-reversible plating/stripping of K was demonstrated independently of any electrode surface treatment or coating (reference 29 in the manuscript, *Energy Storage Materials* 45, 291) and, in 2022, the solvation number of this electrolyte and its contribution to the thermodynamical stability was determined (reference 42 in the manuscript, *J. Am. Chem. Soc.* 139, 9475). Considering all the above, the authors claim regarding the novelty of their work should be revised and put into context.

From these to works cited by the authors, one can conclude that the thermodynamic stability is better for high electrolyte concentrations. However, in this manuscript, as per Figure 3b, the ionic conductivity of KFSI:DME seems to reach a maximum (inflexion point) at 2.0 m whilst the thermodynamic factor increases after ~ 0.5 m (Figure 4b). Is this the same conclusion obtained in previous papers (29, 42) that establish the best stability at high concentrations? Needless to say that the way to express concentration across the mentioned works is different so a unique criteria should be established for the comparison.

Regarding the XPS peak interpretation, there are several points that have to be profoundly revised:

- the authors assign the C 1s peak found at 282.7 eV (Figure 2c) to a non-identified carbide. Indeed, at those binding energies, transition-metal carbides should appear and, in that case, the 2p, 3d or 4f photoelectron peak corresponding to the transition metal should also be present. However, in the survey spectra there are no traces of transition metals.

- in Figures 2c and 2d, the authors assign to a carbonate species (CO_3^{2-}) the O 1s peak at around 530 eV. If this was the case, the corresponding component should be seen with an intensity of 1/2.9 times that of the oxygen component. Please note that if XPS sensitivity factor is normalized to carbon 1s, then for oxygen 1s is 2.9. In fact, the presence of Li_2CO_3 , Na_2CO_3 , K_2CO_3 ,... corresponds with a peak at >531.5 eV (*ACS Appl. Mater. Interfaces*, 7, 7801).

The presence of the Na KLL Auger transition in the O 1s region is not ideal since it masks the identification of species. The impact of this overlap can be evaluated by using an Mg X-ray source (ACS Appl. Mater. Interfaces, 7, 7801) so the Na KLL peak is shifted in binding energy down to ~264 eV, out of the way of regions of interest with some marginal overlap with the K 2p peak. In this case, the F KLL peak could be seen close to the F 1s and could be used to establish an additional binding energy calibration.

At this stage, despite the undoubted interest of this research and the careful analysis of the experimental data presented, I cannot recommend the publication of this paper in Nature Communications.

Reviewer #2 (Remarks to the Author):

This paper deals with the characterization of the ionic transport and thermodynamic properties of the KFSI electrolyte solutions. Basically, standard measurements are performed for the study of the KFSI electrolyte in this work. The authors particularly introduce a preparation protocol for K metal electrodes with higher purity and homogeneous surface for accurate measurements. Most of the results are presented in sufficient detail and technically sound. However, the significance of the results is insufficient for publication in nature communication. Some interpretations are inconsistent.

1) The authors mentioned that “the KFSI in DME is used as a model electrolyte system because of the ability of the FSI-anion to more surface effectively passivate the K metal”(page 2). Fig.2d indicates only KOH on the pure K electrode forms, which likely originates from the XPS measurement. These results are used as evidence for the improved K metal stability enabled by the metal preparation protocol. However, in the discussion of the Hittorf experiments, the authors also state “the high reactivity of K results in continuous formation of SEI, shown by the increasing impedance in Supplementary Fig. 20 for KFSI:DME” (page 5). In fact, Supplementary Fig. 9 indicates that there could be some F species on the K surface.

2) Supplementary Fig. 20 shows the increasing impedance of the K symmetric cells with KFSI electrolytes. It seems that the impedance of the cells for the Hittorf experiments is not yet stabilized within a rest time of 4 h. Does the impedance influence the results of the measurements?

3) The authors claim that there could be side reactions or dendrite formation in the cathodic chamber. Is it possible to tune the experiment conditions (current, polarisation time) to overcome these issues?

Reviewer #3 (Remarks to the Author):

In this work, the authors reported their results on characterising the ionic transport and thermodynamic properties of a nonaqueous K-ion electrolyte. The results are interesting and valuable to the development of K-ion batteries. However, this work is too specific in my opinion can be of interest for the limited community of electrochemists not for the wider audience. This paper should be transferred to one of electrochemical journals such as Journal of Power Sources. Publications in the journal like Nature Communications should address to the wide scientific society. The novelty is also not high being published in Nature Communication. Therefore, I do not recommend publication in Nature Communications.

Response to the Reviewers' Comments

We would like to thank all the reviewers for their constructive and insightful comments. Changes implemented to the original document are highlighted in yellow for the direct quote to the revised manuscript.

There seems to be an underlying apprehension about the impact of our study and its fit for *Nature Communications* that we would like to address upfront. This is the first time that all the four key transport and thermodynamic properties (the transference number, thermodynamic factor, ionic conductivity, and salt diffusivity) have been fully characterised for a K-ion electrolyte. Knowledge of all these properties is required to accurately model the electrochemical behaviour of any battery chemistry with the Doyle-Fuller-Newman (DFN) model. There has been speculation of the potential of K-ion to be a faster charging chemistry compared to Li-ion, however, until our study such claims have only been speculative.

Upon reviewing the manuscript in light of the reviewers' comments, we recognize that we did not effectively convey and demonstrate the broader impact of our research. We have therefore decided to conduct Doyle-Fuller-Newman (DFN) modelling of the charging performance of a graphite half-cell for Li-ion and K-ion to illustrate the impact of these “faster” K-ion electrolyte transport properties on battery fast-charging capability. We use the empirical transport and thermodynamic property relationships of the KFSI and LiFSI electrolytes characterised in this paper (Supplementary Note 9) and the battery modelling package PyBaMM to conduct the simulations.

Since graphite is the leading anode for both Li-ion and K-ion and limitation in achieving fast charging in Li-ion, the graphite half-cell was deemed the most appropriate comparison. The results are reported in **Figure 6** in the revised manuscript and is shown below.

The results show significantly greater accessible capacity at high-rates and reduced concentration gradient formation for the K-ion cell compared to the Li-ion cell. We believe these results more clearly demonstrate the significance of the faster transport properties of the K electrolyte indicating the potential of K-ion batteries for high-power fast charging, a finding we believe will be of significant interest to a wider scientific audience.

The full model details are described in the Methods section and the results and discussion are in the main text. We have added this as a new “Modelling” results section and Figure 6 in the main text.

Fig. 6. Doyle-Fuller-Newman (DFN) simulations of a K-ion and Li-ion graphite half-cell during charge with KFSI and LiFSI in DME electrolytes using the transport and thermodynamic property relationships characterised (Supplementary Note 9). Modelled using PyBaMM.⁵⁵ (a) Simulated graphite half-cells after 5 minutes of 2C constant current charge including Li^+/K^+ electrolyte concentration gradient. (b) Li^+/K^+ electrolyte concentration gradient over time during 2C charge (c) Concentration overpotentials for the half-cells with increasing C-rate. (d) Accessible capacity for the half-cells with increasing C-rate.

Results

Modelling

To understand the impact of the differences in the electrolyte transport and thermodynamic properties on cell performance, we conducted Doyle-Fuller-Newman (DFN)^{17,56} simulations of the charging behaviour of K-ion and Li-ion graphite half-cells with KFSI and LiFSI:DME electrolytes. The four properties characterised here (D , t_0^+ , κ and χ_M) are all the electrolyte properties required for DFN cell modelling, therefore, this is the first time K-ion has been simulated using the DFN model. The cells were modelled using the battery modelling package PyBaMM⁵⁵ in combination with the empirical transport and thermodynamic property relationships of the KFSI and LiFSI:DME electrolytes characterised in this paper (Supplementary Note 9). Since graphite is the leading anode for both Li-ion and K-ion^{2,4,57} and is the limiting factor in realising fast charging in Li-ion,^{57,58} the graphite half-cell was deemed to be the most appropriate comparison. Full details of the model are described in the Methods.

Figure 6a depicts the half-cell being modelled and shows the significantly reduced electrolyte concentration gradient formation for the KFSI cell compared to the LiFSI cell after 5 minutes of charging at 2C. The LiFSI cell reaches the lower cut-off voltage of 0.01 V very shortly after this due to significant overpotentials, below which Li metal plating would start to occur⁵⁸. Figure 6b shows the concentration gradient formation over time at the same charging rate, again showing reduced electrolyte concentration gradient formation for the KFSI cell and demonstrating the clear advantages of the higher t_0^+ and D for the KFSI electrolyte. Figure 6c shows that the reduced concentration gradient formation results in significantly lower concentration overpotentials (~60% lower for the KFSI compared to the LiFSI cell from 1–3C) again emphasising the importance of t_0^+ and D. Finally, Fig. 6d illustrates the impact on fast charging cell performance, demonstrating that the KFSI cell can achieve significantly higher accessible capacities at higher charging rates (48% vs. 18% at 2C for the KFSI cell compared to the LiFSI cell, respectively), showing superior high-power capability for the KFSI cell. The greater Li-ion concentration overpotentials cause the LiFSI cell to reach the lower cut-off voltage, and hence the Li plating potential, faster than for the KFSI cell, limiting the accessible capacity. From the reduced K^+ electrolyte concentration gradient formation in Fig. 6a and 6b, the electrolyte K^+ concentration at the back of the graphite electrode near the current collector is considerably higher than that for Li^+ , enabling greater accessible K^+ for graphite intercalation and hence greater accessible capacity. Even at the high charging rate of 4C the KFSI cell can access 19% capacity whilst the LiFSI cell only 1%. These results demonstrate the important role of the electrolyte transport properties in increasing high-power performance, particularly t_0^+ and D. The faster electrolyte transport properties combined with the higher potential of K^+ intercalation into graphite,^{2,9} enables superior high-power charging performance of the K-ion chemistry compared to Li-ion.

Nevertheless, this is only a provisional indicative model to demonstrate the potential impact on cell performance of the faster K-ion electrolyte transport properties. Full cell high-power K-ion performance may only be achieved if a suitable electrolyte is developed which provides both a stable SEI for the graphite anode and stability at the high operating voltages of the leading cathodes.² Also for a more accurate and sophisticated full cell model, more advanced characterisation should be conducted of critical electrode properties such as the solid diffusivities and exchange current densities.

Discussion

DFN simulations of K-ion and Li-ion graphite half-cells, using the electrolyte property relationships characterised here, demonstrates the faster transport properties of the KFSI:DME electrolyte results in superior charging rates.

Abstract

Finally, using these characterised properties, we conducted DFN simulations of Li-ion and K-ion graphite half-cells, demonstrating superior K-ion charging rates.

Methods

Modelling

For the DFN modelling, the open-source battery simulation package Python Battery Mathematical Modelling (PyBaMM)⁵⁵ version 23.2 half-cell DFN model and CasADi numerical solver⁶³ was used. For determining the accessible capacity % at different C-rates, the baseline performance was determined from a C/50 charge rate. For the KFSI and LiFSI:DME electrolytes the empirical concentration dependent functions characterised in this paper were used (Supplementary Note 9). The 'Ai2020' pouch cell parameter set was used for the Li graphite and cell geometry base parameters.⁶⁴ The default half-cell metal electrode parameters were used⁶⁵ with a metal electrode thickness of 12.5 μm . Between the Li-ion and K-ion graphite, electrode parameters such as particle size, porosity, and graphite electrode geometries were kept constant as defined in the parameter set.⁶⁴ The graphite diffusivity function was also kept constant since the K graphite diffusivities characterised so far are similar to that of Li.^{17,66,67} The exchange current density function was also kept constant as no K^+ graphite exchange current density has yet been characterised. We incorporated the K graphite OCV profile for the K graphite simulation based on characterised data.⁶⁸ The lower voltage cut-off was set as 0.01 V vs. K^+/K or Li^+/Li below which metal plating would occur.

Reviewer #1

The authors have submitted the paper “Characterising the Ionic Transport and Thermodynamic Properties of Potassium-ion Electrolytes” which is a comparative study of the non-aqueous K-ion electrolyte KFSI:DME with the well-known Li-ion counterpart LiFSI:DME.

Indeed, the understanding of the ionic transport and thermodynamic properties of LiFSI:DME is crucial for the further development of K-ion batteries.

We thank the reviewer for their constructive comments.

The authors claim that “there is currently no study which has fully characterised the ionic transport and thermodynamic properties for nonaqueous K-ion electrolytes.” Back in 2017, the long-term and highly-reversible plating/stripping of K was demonstrated independently of any electrode surface treatment or coating (reference 29 in the manuscript, Energy Storage Materials 45, 291) and, in 2022, the solvation number of this electrolyte and its contribution to the thermodynamical stability was determined (reference 42 in the manuscript, J. Am. Chem. Soc. 139, 9475). Considering all the above, the authors claim regarding the novelty of their work should be revised and put into context.

The electrolyte ionic transport and thermodynamic properties as defined in the main text refer to the transference number, thermodynamic factor, ionic conductivity, and salt diffusivity. Until now no study has fully characterised all four properties for a K-ion electrolyte.

Reference 29 (now 30) is currently cited as it shows KFSI:DME forms a more uniform SEI with K-metal which is one of the key reasons KFSI:DME was used as a model electrolyte system. Reference 29 (now 30) reports improved electrochemical stability of K metal in KFSI:DME electrolyte, without surface treatment or coating. However, the authors did not characterise the electrolyte. Moreover, the stability achieved is still insufficient for electrolyte transport characterisation, for instance for characterising the salt diffusivity using the most accurate technique: restricted diffusion which requires an exponential relaxation of the OCV. We demonstrate this in **Supplementary Fig. 1** where, using standard K preparation in KFSI:DME, the salt diffusivity cannot be determined due to the erratic OCV behaviour whereas our preparation results in the required exponential OCV relaxation to determine the salt diffusivity (**Supplementary Fig. 13**). Thus, despite the fact this reference achieved reversible plating/stripping of K which we acknowledge, the stability achieved is insufficient for K electrolyte characterisation.

Reference 42 (now 33) is currently cited for its important contribution to the understanding of KFSI:DME solvation structure which support our thermodynamic factor results. However, again it does not characterise any of the ionic transport or thermodynamic properties of the electrolyte that have been characterised here, which is the point and novelty of our paper.

Since these are important papers on this electrolyte, we have emphasised their contributions more clearly, providing additional context in the main text.

Introduction

However, there is currently no study which has fully characterised the ionic transport and thermodynamic properties (D , t_0^+ , κ and χ_M) for a K-ion electrolyte.

KFSI:DME has been shown to enable reversible plating and stripping of K metal due to formation of a more uniform SEI, indicating an appropriate model electrolyte for the electrochemical measurements which require symmetric metal cells.³⁰ Moreover, Le Pham et al have investigated the solvation structure of K^+ in KFSI:DME electrolytes using operando XRD and Raman spectroscopy.³³

From these to works cited by the authors, one can conclude that the thermodynamic stability is better for high electrolyte concentrations. However, in this manuscript, as per Figure 3b, the ionic conductivity of KFSI:DME seems to reach a maximum (inflexion point) at 2.0 m whilst the thermodynamic factor increases after ~ 0.5 m (Figure 4b). Is this the same conclusion obtained in previous papers (29, 42) that establish the best stability at high concentrations? Needless to say that the way to express concentration across the mentioned works is different so a unique criteria should be established for the comparison.

To confirm, there is no contradiction with the conclusions of these papers. In reference 29 (now 30) the authors determine that the electrochemical stability (measured by assessing the electrochemical stability window using linear sweep voltammetry) increases with concentration. However, this electrochemical stability is not being assessed by our work.

The thermodynamic factor is not a measure of stability, it is instead a measure of ideality and deviation from Nernstian behaviour. The thermodynamic factor is defined as (Eq. 2 in manuscript):

$$\chi_M = 1 + \frac{d \ln(f_{\pm})}{d \ln(c)}$$

The ionic conductivity reaching a maximum and the thermodynamic factor increasing with increasing concentration is the standard observation for binary electrolytes (references 15, 18, 20, 26, 27, 40). As explained in the paper the ionic conductivity reaches a maximum due to increasing ion-ion and solvent interactions, coupled with the increasing viscosity. Meanwhile with increasing concentration the thermodynamic factor increases due to greater ion-solvent interactions, resulting in DME being increasingly bound, reducing solvent vapour pressure, and thus increasing the salt activity coefficient and the resulting thermodynamic factor. These results are consistent with other electrolytes characterised (references 15, 18, 20, 26, 27, 40).

In reference 42 (now 33) the reduced solvation numbers in highly concentrated (> 5 M) KFSI in DME [N.B. this is significantly greater than the concentration range of our work i.e. 0.25–2 m = ~ 0.2 –1.5 M] results in a thermodynamically more favoured K-ion desolvation at the graphite surface thus preventing the solvent co-intercalation phenomenon observed at lower concentrations. Again, this conclusion does not in any way contradict our findings. In fact, their findings on the solvation number supports our thermodynamic factor results and we cite this in

the thermodynamic factor results section. We are not assessing solvation/desorption, stability or co-intercalation, but instead ion transport in the electrolyte.

Regarding the XPS peak interpretation, there are several points that have to be profoundly revised:

We thank the reviewer for their comments on XPS peak interpretation. Upon re-examination of our XPS spectra we discovered that we did make a mistake in the assignment of one of the peaks. We originally assigned the peak at approximately 527 eV in **Fig. 2c and 2d** to an oxide O 1s peak, however, this is actually an additional Na KLL Auger peak (**Fig. R2**). This has now been corrected.

- the authors assign the C 1s peak found at 282.7 eV (Figure 2c) to a non-identified carbide. Indeed, at those binding energies, transition-metal carbides should appear and, in that case, the 2p, 3d or 4f photoelectron peak corresponding to the transition metal should also be present. However, in the survey spectra there are no traces of transition metals.

We agree with the reviewer that the origin of the low binding energy C 1s peak cannot be conclusively identified from the XPS data presented. We have therefore performed additional XPS experiments to uncover the origin of this peak. **Figure R1** presents the XPS spectra obtained during Ar⁺ depth profiling (4 kV, 3 mm × 3 mm) for a total of 5 minutes. It is evident from the C 1s spectra in **Fig. R1b** that this peak once again emerges, appearing as a shoulder on the -CH_x- peak after 1 minute of Ar⁺ sputtering, and then increasing in intensity with increasing sputtering time. The only other peak that appears in the survey spectra during this period (**Fig. R1a**) is the small peak at a binding energy of approximately 240 eV. Core-level spectra around this energy (**Fig. R1c**) reveal the emergence of a doublet peak that is consistent with Ar 2p (Surface Science Spectra, 1992, 1, 376), indicating that argon implantation is taking place at the surface.

Fig. R1. XPS depth profiles of literature preparation K metal electrode to determine the origin of the carbide species (a) survey (b) K 2p, C 1s (c) Ar 2p.

We have not detected any peaks corresponding to transition metals, so we conclude that a transition metal carbide cannot be the responsible species. We therefore suspect that the C 1s peak is instead due to either a potassium or sodium acetylide species. Although there are few reports on these materials, the presence of Li_2C_2 has been confirmed in battery grade lithium metal (J. Power Sources, 2012, 217, 98-101) and it generates a C1s peak at a binding energy similar to that observed here (J. Electrochem. Soc., 1994, 141, 2379). It has been speculated that Li_2C_2 is present as a bulk impurity from lithium production (J. Power Sources, 2012, 217, 98-101), or that it forms due to a reaction between metallic lithium and organic matter (ACS Appl. Energy Mater., 2018, 2, 873-881). It is probable that similar processes could occur for sodium or potassium, producing a carbide species that is responsible for the observed peak.

Alternatively, a potassium or sodium carbide species could form during the high-energy Ar^+ sputtering process. Argon-ion bombardment-induced mixing has been reported to induce carbide formation during the sputtering of silicon (J. App. Phys., 1996, 79, 2934) and titanium (Nucl. Instrum. Methods. Phys. Res. B, 2001, 182, 218-226). In this case, surface carbon species could be mixed with sodium or potassium compounds during argon sputtering, promoting the formation of a sodium or potassium acetylide.

We have added **Fig. R1** to the Supplementary Information as **Supplementary Fig. 18** and have included this discussion on the potential origin of the carbide peak in Supplementary Note 3. We have also updated **Fig. 2c** to acknowledge the presence of this carbide species.

Fig. 2c. XPS depth profiles on K metal after 5, 15 and 25 min of Ar^+ sputtering (c) O 1s, K 2p and C 1s spectra from the standard preparation.

- in Figures 2c and 2d, the authors assign to a carbonate species (CO_3^{2-}) the O 1s peak at around 530 eV. If this was the case, the corresponding component should be seen with an

intensity of 1/2.9 times that of the oxygen component. Please note that if XPS sensitivity factor is normalized to carbon 1s, then for oxygen 1s is 2.9. In fact, the presence of Li₂CO₃, Na₂CO₃, K₂CO₃,... corresponds with a peak at >531.5 eV (ACS Appl. Mater. Interfaces, 7, 7801).

As the reviewer correctly highlights, the Scofield relative sensitivity factor (RSF) for O 1s is 2.93, normalised to C 1s. In a carbonate species there are three oxygen atoms for every one carbon atom, and as such, the expected intensity of the C 1s peak is $1/(3 \times 2.93) \approx 0.114$ times that of the corresponding O 1s peak. For example, the O 1s carbonate peak after 5 minutes of Ar⁺ sputtering in **Fig. 2c** has a raw peak area of 2527 CPSeV, so the corresponding carbon 1s peak should have a raw area of roughly 287 CPSeV. This is a relatively low peak intensity and, although the existence of a carbonate C 1s peak can be clearly seen by eye in **Fig. 2c**, it is not possible to accurately determine a peak area due to the low intensity-to-background ratio and the overlap with the potassium 2p_{3/2} peak. Nevertheless, several attempts to quantify the intensity of this peak resulted in a rough raw area of 250 CPSeV, which is consistent with our assignment to a carbonate species.

Additionally, the carbonate O 1s and C 1s raw peak areas after 5 minutes of Ar⁺ sputtering in **Fig. 2d** are 4239 and 270 CPSeV, respectively. This gives an approximate carbon:oxygen ratio of 1:5.4, which is more oxygen-rich than the theoretical value of 1:3. However, accurate quantitative analysis is again hindered by the presence of the high intensity K 2p_{3/2} peaks, and it should be noted that recent measurements of K₂CO₃ reference materials using XPS have also appeared slightly oxygen rich (references 36 and 37).

We further recognise the reviewer's comments that carbonate species typically correspond to O 1s peaks with binding energies above 531.5 eV. This is true for most metal carbonates, including Li₂CO₃ with an O 1s binding energy of 532 eV (reference 36). However, sodium and potassium have lower electronegativities than lithium, so their valence electrons are drawn towards the oxygen atoms in Na₂CO₃ and K₂CO₃, increasing the shielding between the oxygen nucleus and the 1s electrons and reducing their binding energy. Na₂CO₃ O 1s binding energies have therefore been reported in the range of 530.8–531.4 eV (ACS Appl. Mater. Interfaces, 2015, 7, 7801 and reference 36) and K₂CO₃ O 1s binding energies in the range of 530.5–531 eV are common in the literature (references 36 and 37). These reported binding energies are consistent with our assigned carbonate peak positions. We also acknowledge that absolute peak positions are strongly influenced by the XPS instrument and measurement conditions, so we would like to highlight that the relative positions of our assigned carbonate and hydroxide peaks show excellent agreement with the literature (Fig. 4 and S2 of reference 36). We are therefore confident that we have made an accurate peak assignment.

The presence of the Na KLL Auger transition in the O 1s region is not ideal since it masks the identification of species. The impact of this overlap can be evaluated by using an Mg X-ray source (ACS Appl. Mater. Interfaces, 7, 7801) so the Na KLL peak is shifted in binding energy down to ~264 eV, out of the way of regions of interest with some marginal overlap with the K 2p peak. In this case, the F KLL peak could be seen close to the F 1s and could be used to establish an additional binding energy calibration.

We agree with the reviewer that the Na KLL Auger transition complicates the accurate quantification of species in the O 1s region. However, we do not believe that this prevents us from drawing meaningful qualitative conclusions from the XPS data we have presented. For example, the O 1s peaks in **Fig. 2d** have limited overlap with the Na KLL Auger transitions, revealing a high-intensity hydroxide peak. Clear differences can be observed between the relative intensities of the hydroxide peak between the two samples, even when considering potential overlap with the Na KLL peaks.

We thank the reviewer for the useful suggestion of using an alternative X-ray source. We performed an additional XPS measurement on a Na metal reference sample (Al X-ray source) to determine the extent of the Na KLL Auger transitions (**Fig. R2**), revealing that they result in peaks at binding energies from ~490–565 eV in our system. A Mg source (1253.6 eV) would shift the apparent binding energies of the Na KLL Auger transitions down by 233 eV relative to the Al source (1486.6 eV) we utilised in this study. This would bring the Na KLL Auger transitions down to ~257–332 eV, with the main Na KLL Auger transition appearing at an apparent binding energy of approximately 264 eV, which could interfere with the potassium 2p and carbon 1s peaks, as the reviewer mentioned. This overlap could make it challenging to fit a background to the K 2p and C 1s regions, complicating peak fitting and the accurate quantification of peak areas once more.

Fig. R2. Na KLL Auger spectrum, measured from the surface of a Na metal reference sample. The O 1s peaks at ~531 eV are consistent with Na₂CO₃ and NaOH.

We further recognise the value in establishing additional binding energy calibrations. Unfortunately, fluorine is only present as a trace impurity in the potassium samples we investigated (**Supplementary Fig. 3 and 5**) and would not produce peaks with sufficient intensity to act as a suitable standard. The use of a Mg X-ray source could therefore be particularly helpful in alternative studies on the solid electrolyte interphase (SEI) that forms on K metal, for example, where the fluorine content is likely to be much higher.

At this stage, despite the undoubted interest of this research and the careful analysis of the experimental data presented, I cannot recommend the publication of this paper in Nature Communications.

We hope to have now addressed all the reviewer's concerns.

Reviewer #2

This paper deals with the characterization of the ionic transport and thermodynamic properties of the KFSI electrolyte solutions. Basically, standard measurements are performed for the study of the KFSI electrolyte in this work. The authors particularly introduce a preparation protocol for K metal electrodes with higher purity and homogeneous surface for accurate measurements. Most of the results are presented in sufficient detail and technically sound. However, the significance of the results is insufficient for publication in nature communication. Some interpretations are inconsistent.

We thank the reviewer for their constructive comments. We hope to have addressed the reviewer's concerns about the impact of our findings in our opening comments, DFN modelling and in the revised manuscript.

1) The authors mentioned that “the KFSI in DME is used as a model electrolyte system because of the ability of the FSI-anion to more surface effectively passivate the K metal”(page 2). Fig.2d indicates only KOH on the pure K electrode forms, which likely originates from the XPS measurement. These results are used as evidence for the improved K metal stability enabled by the metal preparation protocol. However, in the discussion of the Hittorf experiments, the authors also state “the high reactivity of K results in continuous formation of SEI, shown by the increasing impedance in Supplementary Fig. 20 for KFSI:DME” (page 5). In fact, Supplementary Fig. 9 indicates that there could be some F species on the K surface.

We state that we achieve improved and sufficient K metal stability for electrolyte characterisation through our preparation protocol but not complete stability. In fact the lithium metal electrodes used routinely for analogue measurements for Li-ion electrolytes are also not completely stable and continuously form additional SEI during rest in numerous electrolyte systems, despite the lower reactivity compared to metallic potassium (Small, 2020, 16, e2000756, Nat. Energy, 2021, 6, 487–494).

Fortunately, complete stability is not a condition sine qua non to extract accurate and reliable transport and thermodynamic properties. This is clearly demonstrated in **Supplementary Fig. 1** where we compare the restricted diffusion polarisation relaxation profiles of KFSI:DME obtained using K-metal electrodes prepared with the standard procedure and our novel preparation protocol. The exponential relaxation behaviour required for the determination of the salt diffusion coefficient (see main text Diffusion Coefficient section and **Supplementary Fig. 13**) is observed only with K-metal electrodes prepared with our protocol.

The XPS measurements were performed on as-prepared potassium electrodes to assess the composition of the initial potassium surface before any electrolyte was added. We demonstrate that our preparation protocol produces potassium surfaces with reduced levels of impurities at the surface, as can be most clearly seen in **Supplementary Fig. 2a and 4a**. However, there are still some trace impurity species present, including a fluorine-containing species with a F 1s peak at a binding energy of approximately 683 eV which is consistent with KF (Reference 37).

We have stated in our paper that ‘sufficient’ K stability was achieved through our K preparation to undertake the electrolyte characterisation experiments. The challenge with K stability is the reason why no one has fully characterised these properties for a K-ion electrolyte before. We improved the stability sufficiently to undertake these electrochemical characterisation experiments.

2) Supplementary Fig. 20 shows the increasing impedance of the K symmetric cells with KFSI electrolytes. It seems that the impedance of the cells for the Hittorf experiments is not yet stabilized within a rest time of 4 h. Does the impedance influence the results of the measurements?

It is true that the impedance of our symmetric cells does not stabilize during the 4-hour rest period we employed in the Hittorf experiments. However, the changing impedance has no impact on the Hittorf experiment transference results as a constant current was applied for a set duration, ensuring the passing of a controlled quantity of charge, which is independent of impedance.

Figure R3 presents the impedance spectra recorded during the 4-hour rest for the two different 1 m KFSI in DME Hittorf cells. The impedances are different yet the transference number results are exactly the same within the experimental error (**Fig. 3a**), evidencing the fact that the impedance does not affect the results. The true purpose of the initial resting period is to allow the cell to reach equilibrium within the temperature chamber, ensuring that all experiments were performed at $20 \pm 0.3^\circ\text{C}$. We found that a rest time of 4 hours was sufficient for this. As evident in **Fig. R3**, the high-frequency intercept, which corresponds to electrolyte resistance, is initially low and then increases to a stable value of about $1200 \Omega \text{ cm}^2$ (the Hittorf cells are 15 cm long so electrolyte resistance is significant), indicating that the cell has cooled to a stable temperature. Differences in temperature during cell assembly and time between assembly and loading into the Binder Oven are likely responsible for the different initial and final impedances between the two cells.

Figure R3. Impedance spectra taken every hour during the initial 4-hour rest for two different 1 m KFSI in DME Hittorf cells.

3) The authors claim that there could be side reactions or dendrite formation in the cathodic chamber. Is it possible to tune the experiment conditions (current, polarisation time) to overcome these issues?

If on the one hand reducing the current would reduce potential side reactions and non-uniform plating, on the other hand the time required to pass a sufficiently large amount of charge to build a concentration gradient able to produce statistically significant density difference between the chambers increases. Using much longer times would result in the concentration boundary layer extending beyond the anodic and cathodic chamber during the experimental period which would make the measurement void (Equation below) (reference 18, 40 and *Electrochimica Acta*. 167 pp. 357–363):

$$2\sqrt{Dt} < x_{max}$$

Where t is current pulse duration, and x_{max} is the maximum distance that the concentration gradient can progress into the cell.

Based on our diffusivity results, including uncertainty, and the cell geometry, the time can be no longer than 27 hours for high confidence that the concentration boundary layer would not leave the anodic/cathodic chamber. The conditions we used here are an optimum to form a detectable concentration difference with low error whilst preventing the concentration boundary moving out of the anodic and cathodic chambers.

The formation of nonuniform K deposits happens at current densities as low as $10 \mu\text{A cm}^{-2}$ (**Figure R4**, PNAS. 117 11 pp. 5588–5594) and **Fig. R5**, reference 30), which is significantly below the minimum current densities we can use while detecting a statistically significant density difference and maintaining the concentration boundary layer within the chamber. **Fig. R5** shows nonuniform deposition at $50 \mu\text{A cm}^{-2}$ in 5 M KFSI:DME, a concentration significantly higher than any of our concentrations used ($0.25\text{--}2 \text{ m} = \sim 0.2\text{--}1.5 \text{ M}$). In this study, we used a current of $100 \mu\text{A}$ ($127.3 \mu\text{A cm}^{-2}$). We attempted to use lower currents of $75 \mu\text{A}$ and $50 \mu\text{A}$ however the errors increased significantly as the density difference became less statistically significant, therefore, $100 \mu\text{A}$ was deemed the optimum. The inherent challenge for the cathodic chamber is this nonuniform plating and this can be clearly seen in our photograph of the cathodic Hittorf electrode in 1 m KFSI:DME shown in **Fig. R6**. Nonuniform K/Li deposition is the likely reason for the discolouration in both the Li and K cathodic chambers (**Supplementary Fig. 9**) as mossy Li/K can form during plating and become detached from the electrode (dead metal). These deposits have a large surface area and can result in significant formation of new SEI and consumption of the electrolyte species, hence the exaggerated reduction in density and concentration as explained in the main text. This effect is more severe for K given its much greater reactivity. The discounting of cathodic data has also been reported in other studies evaluating Li electrolytes with the densitometric Hittorf method for the same reasons due to cathodic dendrites/side reactions (reference 18). In the anodic chamber where stripping occurs this is not an issue. We have amended the explanation in the main text to be clearer.

We utilised atomic force microscopy (AFM) to examine the topography of our pristine K metal electrodes before the addition of electrolyte, which we have also included in the main text (**Fig. 2b**). As evident in **Fig. R7**, the surface is initially smooth and uniform, with evidence of shallow grooves introduced by the microtome polishing process. The changes in surface structure evident in **Fig. R6** must therefore be a result of the nonuniform plating process. The roughness after plating is too significant to measure a comparison AFM height map.

Figure R4. SEM image of K metal surface in K-K symmetric cells cycled at 10 μA/cm² in 0.8 KPF₆ in DME showing large nonuniform deposits on the surface (PNAS. 117 11 pp. 5588–5594).

Figure R5. SEM image of plated K with 5 M KFSI:DME at 50 μA/cm² showing nonuniform deposition on the surface (reference 30).

Figure R6. Cathodic Hittorf K electrode after 1m KFSI:DME Hittorf experiment showing significant nonuniform plating

Figure R7. AFM height map of our pristine K metal electrode (scale bar, 20 μm). Inset of Figure 2b

Results

Transference number

Supplementary Fig. 7 shows the anodic and cathodic chamber t_0^+ measurements, with $t_{0\text{K}^+}$ exhibiting significant deviation between them. The cathodic data was discounted due to evidence of nonuniform K deposition and cathodic electrolyte discoloration in both systems (Supplementary Fig. 9). These high surface area K metal deposits result in the exaggerated, continuous formation of SEI and significant electrolyte consumption. Therefore, given the highly sensitive density measurements, the cathodic transference numbers are likely underestimated. There is evidence in the literature of nonuniform K deposition in the same electrolyte at much higher concentrations (5 M) and lower current densities than can be used in our investigation (Supplementary Note 4).³⁰ Significant discoloration was also observed in the cathodic solution for many of the Li- and K-ion experiments, further supporting the nonuniform Li/K deposition as mossy Li/K formed during plating can become detached from the electrode, a phenomenon known in the field as “dead” metal¹¹. Similar discoloration was found in the study by Hou and Monroe using metallic lithium and their cathodic data was also discounted.¹⁸

Reviewer #3

In this work, the authors reported their results on characterising the ionic transport and thermodynamic properties of a nonaqueous K-ion electrolyte. The results are interesting and valuable to the development of K-ion batteries.

We thank the reviewer for their comments.

However, this work is too specific in my opinion can be of interest for the limited community of electrochemists not for the wider audience. This paper should be transferred to one of electrochemical journals such as Journal of Power Sources. Publications in the journal like Nature Communications should address to the wide scientific society. The novelty is also not high being published in Nature Communication. Therefore, I do not recommend publication in Nature Communications.

We hope in light of the new modelling conducted which demonstrates the impact of the faster K electrolyte transport properties on realising superior high-rate cell performance will appeal to the wider scientific community. Please see the additional section and Figure on Modelling and the general statement above providing a brief overview of the model results.

REVIEWERS' COMMENTS

Reviewer #1 (Remarks to the Author):

The authors have carefully revised the manuscript while answering all questions and requests.

At the same time, the authors have better explained the interest and scope of their work.

I would like to apologize since I have realized that the C/O RSF ratio that I mentioned in one of my questions did not consider the number of atoms in the molecule and that could have resulted in some confusion for the authors. In any case they have properly considered the comment delivering a satisfactory answer.

I recommend publication in Nature Communications.

Reviewer #2 (Remarks to the Author):

The modifications made in the revised manuscript did clarify most of the points raised by the reviewer. However, considering the technical content of this paper is specific to the analysis and properties of the KFSI-DME electrolyte, I regret that I cannot recommend the publication of this paper in Nature Communications.